# Biomarker Panel for the Diagnosis of Pancreatic Ductal Adenocarcinoma

**DOI:** 10.3390/cancers12061443

**Published:** 2020-06-01

**Authors:** Hongbeom Kim, Kyung Nam Kang, Yong Sung Shin, Yoonhyeong Byun, Youngmin Han, Wooil Kwon, Chul Woo Kim, Jin-Young Jang

**Affiliations:** 1Departments of Surgery and Cancer Research Institute, Seoul National University College of Medicine, Seoul 03080, Korea; surgeonkhb@gmail.com (H.K.); yoonhyung9@naver.com (Y.B.); vickijoa@naver.com (Y.H.); willdoc78@gmail.com (W.K.); 2BIOINFRA Life Science Inc., Seoul 03127, Korea; kyungnam.kang@bioinfra.co.kr (K.N.K.); yongsung.shin@bioinfra.co.kr (Y.S.S.); chulwoo.kim@bioinfra.co.kr (C.W.K.)

**Keywords:** pancreas, cancer, screening, diagnosis, biomarker

## Abstract

A single tumor marker has a low diagnostic value in pancreatic cancer. Combinations of multiple biomarkers and unique analysis algorithms can be applied to overcome these limitations. This study sought to develop diagnostic algorithms using multiple biomarker panels and to validate their performance in the diagnosis of pancreatic ductal adenocarcinoma (PDAC). We used blood samples from 180 PDAC patients and 573 healthy controls. Candidate markers consisted of 11 markers that are commonly expressed in various cancers and which have previously demonstrated increased expression in pancreatic cancer. Samples were divided into training and validation sets. Five linear or non-linear classification methods were used to determine the optimal model. Differences were identified in 10 out of the 11 markers tested. We identified 2047 combinations, all of which were applied to 5 separate algorithms. The new biomarker combination consisted of 6 markers (ApoA1, CA125, CA19-9, CEA, ApoA2, and TTR). The area under the curve, specificity, and sensitivity were 0.992, 95%, and 96%, respectively, in the training set. Meanwhile, the measures were 0.993, 96%, and 93% in the validation set. This study demonstrated the utility of multiple biomarker combinations in the early detection of PDAC. A diagnostic panel of 6 biomarkers was developed and validated. These algorithms will assist in the early diagnosis of PDAC.

## 1. Introduction

Pancreatic cancer is the third leading cause of cancer-related death. In the future, it is expected to become the second leading cause of cancer-related death, following lung cancer [1]. Despite advances in surgical techniques and the introduction of new treatment strategies, the prognosis of pancreatic cancer remains poor [2]. The poor prognosis is mainly because patients frequently present in an inoperable metastatic state or locally advanced state at the time of diagnosis. There are no pancreatic cancer-specific symptoms; therefore, early detection is difficult. Only 20% of all patients with pancreatic cancer are eligible for surgery [3]. Early diagnosis is required to improve pancreatic cancer survival. 

Pancreatic cancer is diagnosed through imaging technologies such as computed tomography (CT) or magnetic resonance imaging (MRI). However, there are many obstacles to adopting these modalities for initial screening. The ideal initial screening test should be efficient, with high sensitivity and specificity, as well as being safe, readily available, convenient, and affordable [4]. Carbohydrate antigen 19-9 (CA19-9) is currently the most effective and widely used biomarker for pancreatic cancer [5,6]. The median diagnostic sensitivity of CA19-9 is 79%, and the median specificity is approximately 80%, limiting the utility of CA19-9 in the screening of pancreatic cancer [6]. The diagnostic value of a single tumor marker is not high in pancreatic cancer. Therefore CA19-9 is more valuable in the detection of recurrence or the assessment of the response to adjuvant treatment [7,8].

Diagnostic methods utilizing combinations of multiple biomarkers can be applied to overcome the limitations of single tumor markers in the screening of pancreatic cancer. The necessity for multiple biomarkers in the diagnosis of pancreatic cancer is due to tumor heterogeneity and the cancer microenvironment. Even among single tumors, differences exist in differentiation or evolutionary steps among intra-tumor cells, resulting in intra-tumor heterogeneity within solid cancers [9]. To comprehensively assess the status of the tumor microenvironment, several markers should be analyzed [10]. An in vitro diagnostic multivariate index assay (IVDMIA), which combines multiple biomarkers and adds unique analysis algorithms, is useful for the diagnosis of cancer [11]. Multiple biomarker panels consisting of 19 serum proteins have previously been constructed via an extensive screening process using serum samples from patients with a variety of cancers as well as healthy controls [12,13,14,15].

The purpose of this study was to develop diagnostic algorithms using multiple biomarker panels and to validate their performance in the diagnoses of pancreatic ductal adenocarcinoma (PDAC). To the knowledge of the authors, this article is the first to evaluate PDAC diagnostic ability in a cancer panel that has already been commercialized and used for various cancer diagnosis.

## 2. Methods

### 2.1. Patient Samples

From July 2010 to May 2015, PDAC samples were collected from patients who underwent surgery with curative-intent at Seoul National University Hospital. Whole blood samples were collected in 10-ml syringes prior to surgery using standard blood collection techniques, and they were stored in EDTA tubes at room temperature for 1 hour. Samples were centrifuged at 3000 g for 5 min, after which supernatants were collected and stored at −80 ℃. Control blood samples were obtained from 573 healthy individuals who visited the hospital for medical check-ups and agreed to participate in the study. Control subjects with confirmed cancer, suspected cancer, or inflammatory conditions that needed medical management were excluded through the following examinations: (1) medical history, (2) physical examination, (3) routine blood analysis, (4) chest X-ray, (5) abdominal sonography or computed tomography, (6) esophagogastroduodenoscopy, (7) colonoscopy, sigmoidoscopy with stool hemoglobin, or computed tomographic colonoscopy, and (8) mammography or breast sonography in women and/or thyroid sonography. Clinico-pathologic demographics and tumor characteristics were obtained for each patient included in this study. The T status, N status, and TNM stage of each tumor were classified according to the 8th edition of the American Joint Committee on Cancer (AJCC) classification. PDAC samples and control samples were assigned randomly to either the training set or the validation set. Two-thirds of the samples were used as the training set, and validation was performed with the remaining one-third of samples. This study was waived from consent. Including waiving informed consent, this study was approved by the Institutional Review Board at Seoul National University Hospital (H-1703-005-835).

### 2.2. Selection of Candidate Markers

The commercial Korean pan-cancer panel consisting of 19 biomarkers has been studied in the context of hepatocellular carcinoma, breast cancer, lung cancer, gastric cancer, colon cancer, and prostate cancer [12,13,14,15]. Of the 19 biomarkers in the panel, 10 markers (Apolipoprotein A1 (ApoA1), cancer antigen 125 (CA125), CA19-9, C-reactive protein (CRP), cytokeratin 19 fragment 21.1 (CYFRA21.1), carcinoembryonic antigen (CEA), ApoA2, transthyretin (TTR), beta-2 microglobulin (B2M), and D.Dimer) were selected, for which an automated immunological and clinical chemistry testing platform was completed. Based on findings reported in the Korean pancreatic cancer diagnostic marker study, leucine-rich alpha-2-glycoprotein 1 (LRG1) was added for a final panel of 11 candidate markers [16].

ApoA1, ApoA2, B2M, CRP, D-Dimer, and TTR were measured on the Cobas c501 (Hoffmann-La Roche AG., Basel, Switzerland) using the immunoturbidimetric method. CA125, CA19-9, CEA, and CYFRA21.1 were measured on the Cobas e601 (Hoffmann-La Roche AG., Basel, Switzerland) using the electrochemiluminescent detection method, according to the manufacturer’s instructions. LRG-1 was measured by an enzyme-linked immunosorbent assay (ELISA) using recombinant LRG1 protein and anti-Human LRG1 antibody (R&D Systems, Minneapolis, MN, USA).

### 2.3. Data Analysis

The Mann–Whitney U test and Wilcoxon rank-sum test were used for the analysis of the 11 candidate protein biomarkers to detect statistically significant differences in biomarker expression between PDAC samples and control samples. A *p*-value of less than 0.01 was considered statistically significant. The data were then log-transformed to minimize the influence of outliers among the biomarker measurements, and data pre-treatment was performed to convert the age data to categorical data to address bias in the distribution of the numerical values for age.

A classification model was generated based on the training data set using linear classification methods (i.e., Generalized Linear Model (GLM) algorithm and Ridge regression algorithm), non-linear classification methods (i.e., Support Vector Machine (SVM) algorithm and Random Forest (RF) algorithm), and a combination of a linear classification method and a non-linear classification method (i.e., the GLM + RF algorithm), which has the advantages of both the linear classification method (i.e., ease of interpretation) and the non-linear classification method (i.e., robust performance). The model was then verified, and its performance was evaluated using a 10-fold cross-validation technique to confirm the stability of the model.

The criteria for evaluating the performance of a classification model are as follows: the area under the curve (AUC) of the receiver operating characteristic (ROC) produced at model generation should be large, and the protein marker panel should demonstrate excellent performance for all 5 of the classification algorithms.

All analysis was performed using the R statistical package (version 3.5.1) and SPSS version 25.0 (IBM SPSS Statistics, Armonk, NY: IBM Corp).

## 3. Results

### 3.1. Clinical Characteristics 

A total of 180 PDAC samples and 573 healthy control samples were included in this study. The mean age of the PDAC patients was 64 years, and 65.0% were male. Pancreaticoduodenectomy, pylorus-preserving or not, was performed in 55.5% of the PDAC cases. Classified according to the AJCC 8th edition, 29.4% were stage 1, 41.1% were stage 2, 16.7% were stage 3, and 12.8% were stage 4. The mean age of the healthy control group was 57 years, and 58.3% were male. The samples were divided into a training data set for selecting optimal marker panels (120 pancreatic cancer samples and 382 normal control samples) and a validation data set for verifying the selected marker panels (60 pancreatic cancer samples and 191 normal control samples). Clinico-pathologic data were balanced evenly between the training and validation sets (Table 1).

### 3.2. Biomarker Selection and Model Development

The overall study process is shown in Figure 1. Among the 11 candidate biomarkers, 10 biomarkers except B2M showed a statistical difference between PDAC and healthy control samples (Figure 2). The marker panels used in the generation of the model consisted of 2047 combinations, which is the total number of all possible combinations (_11_C_1_ + _11_C_2_ + --- + _11_C_11_) of the 11 candidate biomarkers. After adding age and gender variables to each panel, the combination was applied to the five classification algorithms.

Out of the top 10% of the initial 2047 sets, we selected 137 sets containing CEA and CA19-9, as these are used as tumor markers in PDAC and digestive system cancer. The validation data set was then applied to the classification model that had been generated using the selected candidate marker panels to assess whether the model performed similarly for both the validation and training data sets. We selected 32 sets that demonstrated excellent performance and minimal differences between the training and validation sets. Of these, a marker set with excellent performance independent of the linear and non-linear methods was selected as the new marker set. The AUC in the validation set was 0.993 for RF, 0.983 for GLM, 0.986 for GLM + RF, 0.985 for RIDGE, and 0.991 for SVM. The final marker panel consisted of ApoA1, CA125, CA19-9, CEA, ApoA2, and TTR with the RF classification algorithm method.

### 3.3. Diagnostic Performance of New Biomarker Combination Set

The AUC, specificity and sensitivity were 0.992, 95%, and 96% in the training set, and 0.993, 96% and 93% in the validation set. Table 2 shows the diagnostic values when applied to the other statistical algorithms. Comparing CA19-9, CEA, and the combination of CA19-9 + CEA, the diagnostic performance of the new model was excellent.

The AUC of the new model was 0.993 in the validation set, and that of the CEA + CA19-9 combination was 0.960. CEA alone had the lowest diagnostic ability for PDAC, and even when combined with CA19-9, diagnostic performance did not increase (Figure 3). In the validation set, the diagnostic accuracy (sensitivity) was 89% in stage 1, 92% in stage 2, and 100% in stages 3 and 4. Particularly in stages 1 and 2, the new model improved diagnostic ability compared to CA19-9 alone. The diagnostic accuracy (sensitivity) of CA19-9 alone in stages 1 and 2 were 72 % and 83%. However, the diagnostic accuracy of the new model was 89% and 92% in the validation set (Figure 4).

## 4. Discussion

In this study, we identified a combination of 6 biomarkers (ApoA1, CA125, CA19-9, CEA, ApoA2, and TTR) through an RF classification algorithm method that increased the diagnostic accuracy of PDAC to 95%.

In general, a single tumor marker is used to screen for each type of cancer, but the rate of false positives and false negatives is high. Cancer cells do not always secrete tumor markers or do not secret the same tumor marker even within a single tumor. As well, tumor markers may increase in chronic diseases or other cancers [9,17,18]. Diagnostic methods using combinations of multiple biomarkers can be used to overcome the limitation of single tumor marker screening tests. IVDMIA, which combines multiple biomarkers and adds a unique analysis algorithm, is helpful for the diagnosis of cancer [11]. The representative multiple biomarker set, currently used as a diagnostic method in the clinical setting, is Ova1 in ovarian cancer. In September 2009, the FDA approved a serum-based test called OVA1 (Vermillion, Inc., Austin, TX), combining five measured proteins (CA125-II, TTR, ApoA1, B2M, and transferrin) as an adjunct to clinical decision making for women planning surgery for an adnexal mass [19].

There is also a diagnostic antibody microarray platform in pancreatic cancer. This platform, consisting of 29 markers, was able to distinguish patients, with stage I and II PDAC, from controls with a ROC AUC value of 0.96 [20]. However, due to the high cost, it has limited utility as a screening test. In Korea, the multi-marker panel (CA19-9, LRG1, and TTR) that has been developed and validated in large-scale cohorts by multiple reaction monitoring-mass spectrometry (MRM-MS) and immunoassay has clinical applicability in the early detection of PDAC. The triple-marker panel exceeded the diagnostic performance of CA19-9 alone by >10% in all PDAC samples. It was >30% in patients with a normal range of CA19-9. However, an automated system is still being established and has not yet been used in clinical practice [16]. The candidate markers in the present study consisted of 11 markers used in the pan-cancer diagnostic kit, which is commercially available in Korea. This cancer panel can be applied in real clinical practice so that commercialization can progress quickly. This cancer panel is already used for screening seven cancers; hepatocellular carcinoma, breast cancer, lung cancer, gastric cancer, colon cancer, prostate cancer, and ovarian cancer. The sensitivity, specificity, and AUC of the seven cancers are 85–90%, 90–95%, and 0.920–0.992. If pancreatic cancer is included, eight cancers can be screened for 300 USD.

Serum CA19-9 is one of the most widely used serum tumor biomarkers for the detection of PDAC. Serum CEA and CA125 are two other biomarkers that are associated with the tumor burden of PDAC [21,22]. These tumor markers could be applied not only to diagnosis but also to predicting prognosis and assessing treatment response. Xu et al. reported that the combination of postoperative serum CA19-9, CA125, and CEA served to determine a subgroup of patients benefiting from adjuvant chemoradiotherapy [23]. There have been several reports on the relationship between PDAC and Apo. Liu et al. discovered five biomarker combinations that can diagnose PDAC using the MS-based pipeline method, and 3 out of 5 contained Apo (ApoA1, ApoL1, and ApoE) [24]. In the prospective evaluation, when using the combination of ApoA2 isoform (ApoA2-ATQ/AT) and CA19-9, the diagnostic rate was higher than that of CA19-9 alone [25]. ApoA2 was reported to have an essential role in the metastatic process in a study of serum-derived exosomes using iTRAQ-based proteomic analysis [26]. 

There are several reports on the diagnosis of cancers other than pancreatic cancer using the multiple biomarker panel in Korea. This panel was developed from a serum bank containing approximately 4500 samples from 5 types of cancer: breast, colon, stomach, liver, and lung. Kim et al. initially demonstrated the utility of the antibody-bead array approach in identifying signatures specific for primary non-metastatic breast cancer with high accuracy (91.8%) [12]. In non-small cell lung cancer patients, the highest accuracy of multivariate classification algorithms was observed when using the five highest-ranked biomarkers (alpha-1 antitrypsin (A1AT), CYFRA 21.1, insulin-like growth factor (IGF-1), regulated upon activation normal T cell expressed and secreted (RANTES), and alpha-fetoprotein (AFP)). In the validation set, the diagnostic accuracy was 88.2–91.8% according to the analysis algorithm [14]. These multiple panels were also validated in gastrointestinal tract cancer. Ahn et al. identified marker combinations of epidermal growth factor receptor (EGFR), TTR, RANTES, and vitronectin (VN) in gastric cancer with a diagnostic accuracy of 85.9–89.2% [13].

The reason for comparison with single tumor markers is that the tumor markers used in clinical practice or screening of pancreatic cancer are CA19-9 and CEA. Therefore, we tried to show how the diagnostic rate is improved when a new combination of markers is used in actual clinical practice or screening system. Appendix A shows the results of the 2 and 5 marker combination models, the marker combination with the highest discrimination performance seems to outperform the selected panel. However, the reason we did not select the five marker combinations was that they did not fit our selection criteria. The criteria for selecting the optimal marker panel set by the authors was to select a marker panel that showed excellent stability within the combination panel group showing excellent discrimination performance. Stability was evaluated in two ways: (1) The AUC difference between the training and validation set should be small, and (2) the selected panel should show similar stability in 5 classification methods. For this reason, among the 4 and 5 combination marker models, the combination with the highest discrimination performance was judged to have lower stability than the panel, and therefore, was not selected as the optimal marker panel.

In this study, we identified a new combination of protein markers that distinguish between PDAC samples and control samples. This panel has been shown to include markers that were previously not known to be related to pancreatic cancer and demonstrated improved classification performance compared to conventional cancer-specific markers. In addition to identifying new characteristics of previously unknown markers through statistical analysis, this study can be expanded to develop customized models for various purposes such as early diagnosis of pancreatic cancer or developing prognostic models. It can also be used to improve performance by developing complex marker models that combine protein biomarkers with new biomarkers such as DNA and RNA.

This study has limitations. The PDAC patient group contains patients who had surgery for pancreatic cancer. Although 29.5% of all patients were stage 3 or 4, this is a small fraction when considering the staging distribution in the diagnosis of PDAC patients. Additionally, the patients included in this study all received their operations in a single center. Therefore, a large-scale multicenter follow-up study is needed.

## 5. Conclusions

This study demonstrated the utility of a combination of multiple biomarkers for the early detection of PDAC. Diagnostic biomarker panel algorithms that included six biomarkers (ApoA1, CA125, CA19-9, CEA, ApoA2, and TTR) were developed and validated. These algorithms will assist in the diagnosis of early pancreatic cancer, particularly in stage 1 and 2 PDAC. Additional studies with larger cohorts are required to validate these findings.

## Figures and Tables

**Figure 1 cancers-12-01443-f001:**
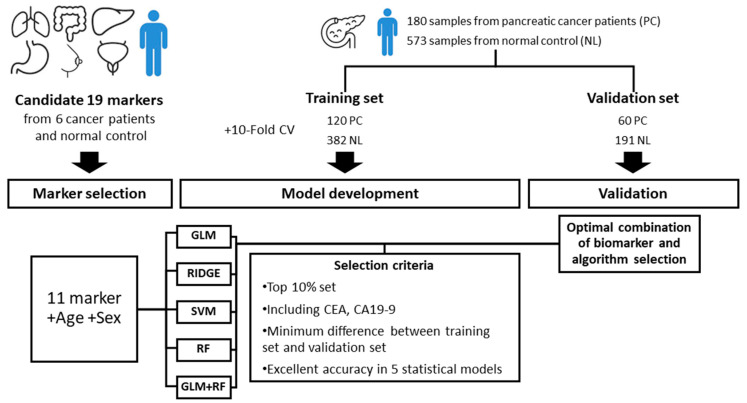
Study schematic flow chart. The marker panels used in the model generation consist of 11 candidate biomarkers and 2047 combinations. After combining age and gender variables to each panel, the combination is applied to the five classification algorithms. Selection criteria for optimal biomarker combinations are as follows; (1) the top 10% out of the 2047 sets, (2) sets containing CEA and CA19-9, (3) minimal difference between the training set and validation set, (4) excellent performance independent of the linear and non-linear methods.

**Figure 2 cancers-12-01443-f002:**
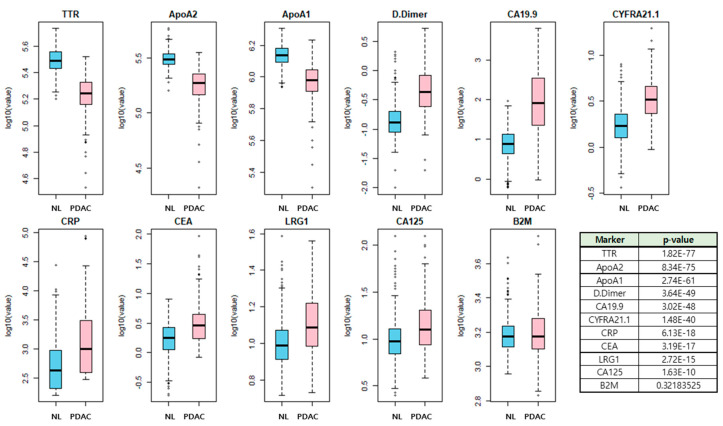
Comparison of 11 candidate markers concentration between PDAC samples and normal control samples. Among the 11 markers (ApoA1, CA125, CA19-9, CRP, CYFRA21.1, LRG1, CEA, ApoA2, TTR, B2M, and D.Dimer), 10 biomarkers except B2M showed statistical differences between PDAC and normal controls.

**Figure 3 cancers-12-01443-f003:**
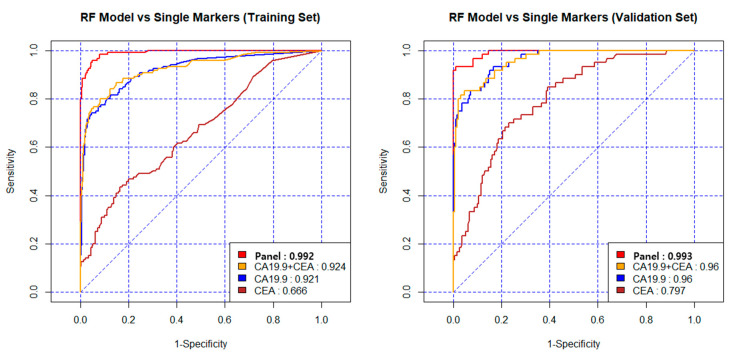
Diagnostic performance of the new model in the training set and validation set. The AUC was 0.992 in the training set and 0.993 in the validation set. Comparing CA19-9, CEA, and the combination of CA19-9 and CEA, the AUC of the new model was 0.993 in the validation set, and the AUC of CEA+ CA19-9 was 0.960 and CA19-9 alone was 0.960.

**Figure 4 cancers-12-01443-f004:**
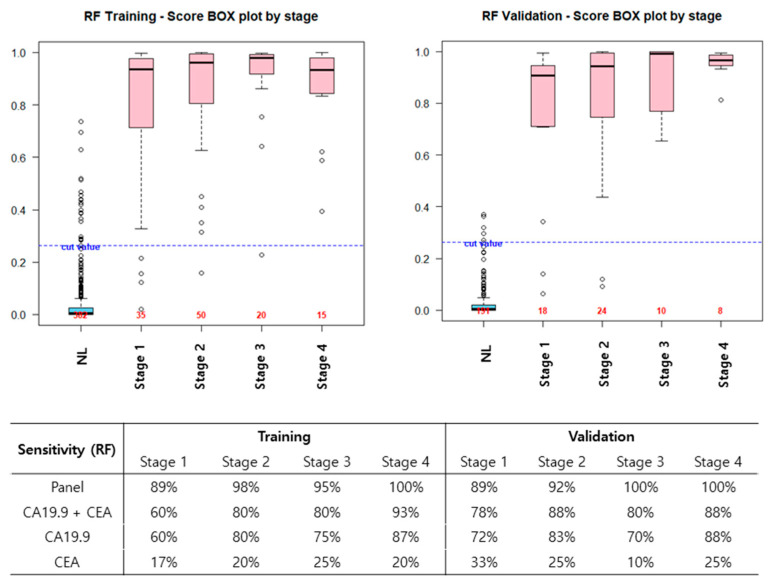
Diagnostic performance, according to the PDAC stage. In the validation set, the diagnostic accuracy, according to the cancer stage was 89%, 92%, 100%, and 100 % in stages 1, 2, 3, and 4, respectively. Notably, in stages 1 and 2, the new model improved the diagnostic ability compared to CA19-9 alone.

**Table 1 cancers-12-01443-t001:** Demographics and clinicopathological characteristics of pancreatic ductal adenocarcinoma patients and healthy controls used in the training and validation sets.

Pancreatic Ductal Adenocarcinoma					
	Total		Training set	Validation set	*p*-value
	*n* = 180		*n* = 120	*n* = 60	
Age		64.4 (9.8)	63.6 (9.9)	66.0 (9.5)	0.109
Sex	M	117 (65.0)	75 (62.5)	42 (70)	0.320
	F	63 (35.0)	45 (37.5)	18 (30)	
Operation	PPPD	62 (34.4)	44 (36.7)	18 (30)	0.996
	PD	38 (21.1)	25 (20.8)	13 (21.7)	
	DP	50 (27.8)	32 (26.7)	18 (30)	
	TP	15 (8.3)	11 (9.2)	4 (6.7)	
	Others *	15 (8.3)	8 (6.7)	7 (11.7)	
Stage	1	53 (29.4)	35 (29.2)	18 (30)	0.996
	2	74 (41.1)	50 (41.7)	24 (40)	
	3	30 (16.7)	20 (16.7)	10 (16.7)	
	4	23 (12.8)	15 (12.5)	8 (13.3)	
T stage	1	22 (12.2)	13 (10.8)	9 (15.0)	0.711
	2	94 (52.2)	66 (55.0)	28 (46.7)	
	3	38 (21.1)	26 (21.7)	12 (20.0)	
	4	10 (5.6)	6 (5.0)	4 (6.7)	
	NA	16 (8.9)	9 (7.5)	7 (11.7)	0.779
N stage	0	69 (38.3)	47 (39.2)	22 (36.7)	
	1	73 (40.6)	47 (39.2)	26 (43.3)	
	2	26 (14.4)	19 (15.8)	7 (11.7)	
	NA	12 (6.7)	7 (5.8)	5 (8.3)	
Differentiation	WD	14 (7.8)	9 (7.5)	5 (8.3)	0.862
	MD	118 (65.6)	80 (66.7)	38 (63.3)	
	PD	26 (14.4)	18 (15.0)	8 (13.3)	
	NA	22 (12.2)	13 (10.8)	9 (15.0)	
Lymphatic	No	78 (43.3)	52 (43.3)	26 (43.3)	0.948
invasion	Yes	77 (42.8)	52 (43.3)	25 (41.7)	
	NA	25 (13.9)	16 (13.3)	9 (15.0)	
Venous	No	61 (33.9)	41 (34.2)	20 (33.3)	0.575
invasion	Yes	85 (47.2)	54 (45.0)	31 (51.7)	
	NA	34 (18.9)	25 (20.8)	9 (15.0)	
Perineural	No	19 (10.6)	14 (11.7)	5 (8.3)	0.547
invasion	Yes	145 (80.6)	97 (80.8)	48 (80.0)	0.547
	NA	16 (8.9)	9 (7.5)	7 (11.7)	
Healthy control					
	Total		Training set	Validation set	*p*-value
	*n* = 573		*n* = 382	*n* = 191	
Age		56.9 (8.8)	56.6 (8.9)	57.5 (8.6)	0.250
Sex	M	334 (58.3)	218 (57.1)	116 (60.7)	0.420
	F	239 (41.7)	164 (42.9)	75 (39.3)	

PPPD, pylorus-preserving pancreaticoduodenectomy; PD, pancreaticoduodenectomy; DP, distal pancreatectomy; TP, total pancreatectomy; NA, not available; WD, well-differentiated; MD, moderated differentiated; PD, poorly differentiated; * Others, bypass surgery and open biopsy.

**Table 2 cancers-12-01443-t002:** Diagnostic performance of new biomarker panel.

Marker	Training and Test Set				Validation Set			
	AUC	Accuracy (%)	Specificity (%)	Sensitivity (%)	AUC	Accuracy (%)	Specificity (%)	Sensitivity (%)
RF								
Panel	0.992	95	95	96	0.993	95	96	93
CA19-9 + CEA	0.924	90	95	76	0.960	92	94	83
CA19-9	0.921	90	95	74	0.960	90	94	78
CEA	0.666	77	95	20	0.797	78	95	25
GLM								
Panel	0.983	94	95	92	0.983	94	95	92
CA19-9 + CEA	0.852	87	95	62	0.928	91	94	80
CA19-9	0.848	88	95	66	0.923	92	94	83
CEA	0.732	78	95	24	0.814	80	96	28
GLM + RF								
Panel	0.984	94	95	92	0.986	95	96	92
CA19-9 + CEA	0.934	91	95	78	0.962	91	93	87
CA19-9	0.933	90	95	75	0.964	90	94	80
CEA	0.732	78	95	24	0.814	80	96	28
RIDGE								
Panel	0.987	95	95	93	0.985	95	96	92
CA19-9 + CEA	0.852	87	95	62	0.928	91	94	80
CA19-9	0.848	88	95	67	0.924	92	94	83
CEA	0.732	78	95	24	0.816	80	96	28
SVM								
Panel	0.990	95	95	95	0.991	97	98	92
CA19-9 + CEA	0.900	89	95	71	0.964	92	97	77
CA19-9	0.912	88	95	68	0.967	92	96	77
CEA	0.627	78	95	25	0.692	78	95	27

AUC, area under the curve; RF, random forest; CA19-9, carbohydrate antigen 19-9; CEA, carcinoembryonic antigen; GLM, generalized linear model; SVM, support vector machine.

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
