# Peer review of "Biomarker Panel for the Diagnosis of Pancreatic Ductal Adenocarcinoma"

_cancers, 2020, doi:10.3390/cancers12061443_

Round 1
Reviewer 1 Report
The article “Biomarker Panel for the Diagnosis of Pancreatic Ductal Adenocarcinoma” by Kim and colab. is interesting and well-written.
I only have one-two suggestions that might improve it slightly.
- In the Patient Samples subheading, rows 69-74, you state that you excluded all patients with confirmed cancer, suspected cancer or inflammatory conditions by means of 8 tests. Am I to understand that all the 573 healthy individuals that were included underwent ALL of these tests?
- I think the authors should place additional emphasis on why their article is important and relevant – at the end of the Introduction section I suggest adding a phrase beginning with “To the knowledge of the authors, this article is the first to...”
- In the discussions section, I think the financial advantages of using this biomarker panel could be discussed more in detail, especially in the current economic context
Minor comments:
- Page 9, rows 195-196 – “but the rate of false positives and false negatives is high.” instead of “but the rate of false positives and false negatives are high.”
Author Response
Response to Reviewer 1 Comments
Point 1
The article “Biomarker Panel for the Diagnosis of Pancreatic Ductal Adenocarcinoma” by Kim and colab. is interesting and well-written.I only have one-two suggestions that might improve it slightly.
In the Patient Samples subheading, rows 69-74, you state that you excluded all patients with confirmed cancer, suspected cancer or inflammatory conditions by means of 8 tests. Am I to understand that all the 573 healthy individuals that were included underwent ALL of these tests?. 

Response 1:
In Korea, health check-up programs are actively being conducted, including the government's health check-up program, where health check-ups are mandatory for people over the age of 40. In addition to basic programs, many people also have various cancer screening at an additional cost. The general cancer screening program includes the above eight tests. Patients included in the study as controls are those who excluded malignancy and inflammatory disease through this test.
Point 2
I think the authors should place additional emphasis on why their article is important and relevant – at the end of the Introduction section I suggest adding a phrase beginning with “To the knowledge of the authors, this article is the first to...” 

Response 2:
Although there have been several studies on pancreatic cancer biomarkers, this study seeks to find new combinations in markers that make up panels that are already commercially available. Therefore, there is an advantage that it can be commercialized immediately. I will add this to the introduction.
Point 3
In the discussions section, I think the financial advantages of using this biomarker panel could be discussed more in detail, especially in the current economic context

Response 3:
Currently commercially available cancer panels can screen 7 cancers; hepatocellular carcinoma, breast cancer, lung cancer, gastric cancer, colon cancer, prostate cancer and ovarian cancer. The sensitivity, specificity and AUC of 7 cancers are 85~90%, 90~95% and 0.920~0.992. If pancreatic cancer is included, 8 cancers can be screened for 300 USD and I think there is a price advantage. I will add this contents to the introduction.
Point 4
Page 9, rows 195-196 – “but the rate of false positives and false negatives is high.” instead of “but the rate of false positives and false negatives are high.”

Response 4:
I'll revise it according to your recommendations

Reviewer 2 Report
This study demonstrated that multiple biomarker panels are useful in the diagnosis of pancreatic ductal adenocarcinoma (PDAC). Among 11 candidate markers, they found that the optimal biomarker combination consisted of 6 markers with high specificity and sensitivity.
Comments.
- You consider 2,047 combinations, which is the total number of all possible combinations of 11 candidate markers. This is a brute-force method and it is quite time-consumption. Have you ever thought about a more efficient way to find potential combinations?
- Although the combination you identified is the optimal biomarker combination in this dataset, it is very likely that it is not an optimal biomarker combination when you apply it to a new dataset. As a result, it is not very meaningful to discuss the optimality problem here.
- Although the study shows that multiple biomarker panels have high accuracy in the diagnosis of PDAC. However, it means you have to collect more biomarker data. Compare with only using one biomarker, your method needs to collect the other five or more biomarkers. It may be more costly to do it but only can increase some accuracy. What is the accuracy if you use less than the 6 biomarkers?
Author Response
Response to Reviewer 2 Comments
Point 1
This study demonstrated that multiple biomarker panels are useful in the diagnosis of pancreatic ductal adenocarcinoma (PDAC). Among 11 candidate markers, they found that the optimal biomarker combination consisted of 6 markers with high specificity and sensitivity.
You consider 2,047 combinations, which is the total number of all possible combinations of 11 candidate markers. This is a brute-force method and it is quite time-consumption. Have you ever thought about a more efficient way to find potential combinations?
Response 1:
The current commercially available panel configuration is 19 markers, and through literature search, 11 of them related to pancreatic cancer were selected and experiments were started. There could be a number of different ways to find the optimal combination of protein marker panels from the 11 protein biomarkers to distinguish pancreatic cancer from normal. Since number of 11 candidate markers is not very high, the authors decided that the best way to find the optimal protein biomarkers was by making 2,047 combinations from all combinable protein marker panels and comparing the performance of those panels as approached in this study. If there were a large number of candidate markers, variable selection methods such as Lasso Regression, Stepwise Selection, and etc. may have been considered to find the optimal panel combination. These methods had been designed due to a time limit of considering all combinations, but with the development of computer hardware, the approach in this study was chosen as it was possible to consider all of 2,047 combinations without such challenge. The time it takes to generate and verify this model is less than 1 hour, although there are some differences in the calculation time for other classification algorithms other than RIDGE which consumes about a day as the calculation time.
Point 2
Although the combination you identified is the optimal biomarker combination in this dataset, it is very likely that it is not an optimal biomarker combination when you apply it to a new dataset. As a result, it is not very meaningful to discuss the optimality problem here.

Response 2:
I agree with your opinion. It is not an optimal biomarker because it started with a limited number of candidate markers and the sample consists of single organ samples. To overcome this, we believe that validation through large-scale multi-center samples is necessary. This is described in the limitation of discussion. We have removed the optimal word from the text that can be misleading.
Point 3
Although the study shows that multiple biomarker panels have high accuracy in the diagnosis of PDAC. However, it means you have to collect more biomarker data. Compare with only using one biomarker, your method needs to collect the other five or more biomarkers. It may be more costly to do it but only can increase some accuracy. What is the accuracy if you use less than the 6 biomarkers?

Response 3:
The reason for comparison with single tumor markers is that the tumor markers used in clinical practice or screening of pancreatic cancer are CA19-9 and CEA. Therefore, we tried to show how the diagnostic rate is improved when a new combination of markers is used in actual clinical practice or screening system.
And we submit additional supplemental tables 1. It is the result of comparing 2 to 5 marker combinations with 6 marker panels among markers in the RF (Random Forest) result list. The comparison method was to find the combination of markers that showed the lowest and highest discrimination performance based on the AUC standard of Training and Validation set within the RF models generated for each combination, and then compared the results.
Looking at the results of the 4 and 5 marker combination models, the marker combination with the highest discrimination performance seems to outperform the selected panel. However, the reason we didn't select the 5 marker combinations was because they didn't fit our selection criteria. The criteria for selecting the optimal marker panel set by the authors was to select marker panel that showed excellent stability within the combination panel group showing excellent discrimination performance. Stability was evaluated in two ways: 1) The AUC difference between training and validation set should be small, and 2) The selected panel should show similar stability in 5 classification methods.
For this reason, among the 4 and 5 combination marker models, the combination with the highest discrimination performance was judged to have lower stability than the panel, and therefore was not selected as the optimal marker panel.
We will summarize this contents and add it to the discussion section.
Round 2
Reviewer 2 Report
The authors has addressed all of my comments. I recommend accepting this paper.